# Women's preference to apply shared decision-making in breast cancer screening: a discrete choice experiment

María José Hernández-Leal ,[1,2,3] María José Pérez-Lacasta ,[1,3] Angels Cardona-Cardona,[4] Núria Codern-Bové ,[5] Carmen Vidal-Lancis ,[6] Montserrat Rue ,[3,7] Carles Forné ,[8,9] Misericòrdia Carles-Lavila ,[1,2,3] on behalf of the Pro-Share Group

## ABSTRACT

**Objective** To analyse women's stated preferences for establishing the relative importance of each attribute of shared decision-making (SDM) and their willingness to pay (WTP) for more participatory care in breast cancer screening programmes (BCSP).

**Design** A discrete choice experiment was designed with 12 questions (choice tasks). It included three attributes: 'How the information is obtained', regarding benefits and harms; whether there is a 'Dialogue for scheduled mammography' between the healthcare professional and the woman; and, 'Who makes the decision', regarding participation in BCSP. Data were obtained using a survey that included 12 choice tasks, 1 question on WTP and 7 socioeconomic-related questions. The analysis was performed using conditional mixed-effect logit regression and stratification according to WTP.

**Setting** Data collection related to BCSP was conducted between June and November 2021 in Catalonia, Spain.

**Participants** Sixty-five women aged between 50 and 60.

**Main outcome measures** Women's perceived utility of each attribute, trade-off on these attributes and WTP for SDM in BCSP.

**Result** The only significant attribute was 'Who makes the decision'. The decision made alone (coefficient=2.879; 95% CI=2.297 to 3.461) and the decision made together with a healthcare professional (2.375; 95% CI=1.573 to 3.177) were the options preferred by women. The former contributes 21% more utility than the latter. Moreover, 52.3% of the women stated a WTP of €10 or more for SDM. Women's preferences regarding attributes did not influence their WTP.

**Conclusions** The participant women refused a current paternalistic model and preferred either SDM or informed decision-making in BCSP.

## STRENGTHS AND LIMITATIONS OF THIS STUDY

⇒ Discrete choice experiments (DCEs) have limitations on the sample size and shared decision-making (SDM), as a little-known model, which makes it difficult for participants to understand the hypothetical situations described in the choice task.

⇒ A DCE is an indirect method to obtain stated preferences and information that otherwise would be impossible to reveal when actual choice behaviour is, in some way, restricted.

⇒ The choice experiments were tested and refined through a two-stage piloting process.

⇒ The responses obtained from participants' willingness to pay may be sensitive when performed within a public health system context.

⇒ This is the first study concerning women's preferences related to SDM in breast cancer screening programmes.

**Correspondence to**
Professor Misericòrdia Carles-Lavila;
misericordia.carles@urv.cat

## BACKGROUND

Breast cancer (BC) screening programmes have been widely adopted in developed countries because of their early detection of malignant lesions, thereby reducing mortality and improving the survival rate.[1 2] In 2020, 34 000 cases were identified and 6600 deaths recorded in Spain due to BC.[3] The screening programme in the Spanish region of Catalonia is scheduled by the Public Health System. This screening programme entails sending a postal letter to women aged between 50 and 69 every 2 years to undergo a free of cost mammography examination in a health centre.[4] Several research and communication campaigns have focused on women's increasing adherence to screening in order to raise awareness of its importance, which in 2017 led the Spanish coverage to reach 81.5% in the target population (although a drop of 73.7% was witnessed 3 years later, during the pandemic).[5] The most cited reason for attending it is the invitation letter issued by the organisations affiliated with the screening programme.[6]

However, the research also recognises the adverse aspects of BC screening, which are usually not explained to women in a balanced manner.[7] These include false positives, false negatives, overdiagnosis and consequently, overtreatment.[8] Overdiagnosis refers to

**Table 1** Attributes and levels of the options for shared decision-making in breast cancer screening

| Attribute | Definition | Model | Description | Levels |
|---|---|---|---|---|
| 1. Information talk | How the information is obtained. | Paternalist | Women are informed via a leaflet on the risks and benefits of BC screening. | Leaflet. |
| | | SDM | Women are informed by a health professional of the risks and benefits of BC screening. | Healthcare professional. |
| 2. Dialogue talk | Dialogue for mammography scheduled. | Paternalist | The health system schedules mammography screening appointments in relation to age criteria. | The health system schedules the mammography. |
| | | SDM | A discussion of therapeutic options and women's values determines the appointment for BC screening. | Women share and discuss their beliefs and values regarding the mammography with a healthcare professional. |
| 3. Decision talk | Who makes the decision. | Paternalist | The decision is made by the healthcare professional. | The healthcare professional makes the decision. |
| | | SDM | The decision is made through a deliberative process between the health professional and the woman. | Shared decision-making by the healthcare professional and the woman. |
| | | IDM | The decision is made exclusively by the woman without the health professional's support. | The woman makes the decision. |

BC, breast cancer; IDM, informed decision-making; SDM, shared decision-making.

screen-detected malignancy that would not have developed into clinical or symptomatic disease and would have never caused health problems throughout the woman's life.[9] Currently, it is impossible to identify which lesions may progress. Therefore, all lesions are generally treated.

In cases of uncertainty, the literature recommends shared decision-making (SDM).[10] SDM is characterised as a participatory care model in which both the professional and the patient are on a par with each other in terms of power to deliberate the best decision. This contrasts with the paternalistic health model, in which the health professional or the system makes such decisions.[11] The latter grew stronger in some health programmes despite declarations and intentions to change it to a more participatory one. SDM allows women, according to their own beliefs and values, to decide together with a health professional whether to undergo a mammography[12] while considering scientific evidence and the options available to them. SDM has shown multiple benefits[13]; despite that, it is necessary to examine its costs, quality and efficiency[13] to determine its applicability in different contexts. Some studies indicate that the application of SDM can mean a reduction in health expenses,[14] whereas other studies suggest that the degree of such savings is unclear.[15 16] Thus, from an economic-health perspective, an approach to cost-benefit analysis can be generated through discrete choice experiments (DCEs).[17]

Using DCE allows to determine the utility of more participatory healthcare in BC screening, enabling women to declare their preferences as well as monetising the intangible values that improve the patient's satisfaction.[18 19]

There is a dearth of studies focusing on awareness of patients' preferences in SDM,[20] and there are none on women's preferences regarding breast cancer screening programmes (BCSPs) and their willingness to pay (WTP) for them. Therefore, this study is aimed to analyse women's declared preferences in Catalonia regarding the attributes of more participatory care (SDM) in contrast to the usual care, which reflects characteristics of a paternalistic model for BC screening; all this by establishing the relative importance of each one through a DCE. In addition, the study participants were asked about their WTP for this type of healthcare.

## METHODS
Our DCE design includes eight hypothetical profiles, referred to as choice sets; each one with three characteristics—attributes—that can present different values or termed levels. For the pairing of these attributes and levels a factorial design is considered, assuming that they are independent from each other. Finally, each *choice set* contrasts with another forming a *choice task*[21] that is presented to women to let them choose as per their preference.

### Definitions of attributes and levels
The three talk model describes and simplifies the implementation of SDM in three steps.[22 23] The attributes and levels of this experiment were defined considering the model and adapting it to the characteristics of the current BCSP. Table 1 describes the levels of a participatory healthcare model with characteristics of SDM (a health professional informs the patient of both risks and benefits and, there is a dialogue between the woman and the healthcare professional (who jointly make the decision) as well as the current model of care, which has characteristics of a paternalistic model (the risks and benefits are provided via a leaflet, the healthcare system schedules

| Conventional choice task (first version) | | |
| --- | --- | --- |
| **Characteristics** | **Option A** | **Option B** |
| Information | I was informed by leaflet | I was informed by a health professional |
| Communication | I discuss mammography with a healthcare professional | The health system schedules my mammography |
| Make a decision | I make the decision of whether to have a mammogram | I make the decision of whether to have a mammogram |
| Choose one option | | |

Pre-test →

| Narrative choice task (final version) | |
| --- | --- |
| **Option A** | **Option B** |
| I receive **information about the benefits and risks in a leaflet**, and I also **talk with a healthcare professional** about my fears or beliefs about mammography. Finally, **I make the decision** to attend -or not- to mammography. | I am **informed by a health professional** about the benefits and risks; my **mammography is scheduled by the health system**, so I do not manifest my fears, beliefs or preferences about the screening to a health professional. Finally, **I make the decision** to attend -or not- to mammography. |

**Figure 1** Examples of a choice task for conventional (first version) and narrative (final version).

the mammography according to age criteria and the healthcare professional makes the decision). Moreover, included in the third attribute ('Decision talk') there is an additional level wherein the woman makes a decision for herself, that is, informed decision-making (IDM). Here, women receive the same information as in the first two attributes, but it is only their responsibility to decide whether to participate in the screening or not.

The current functioning of the BCSP in the Spanish region of Catalonia was used to determine the levels for each attribute: the health system schedules a mammography appointment every 2 years for women aged between 50 and 69 via an invitation letter; an informational leaflet is attached to the invitation letter and, women decide whether to attend it or not. Conversely, in relation to SDM, women inform themselves through a health professional, meeting him/her to discuss their beliefs regarding screening, and decide whether to undergo a mammography.

As a result, two levels were established for both the 'Information talk' and the 'Dialogue talk' attributes representing the functioning of SDM; the typical model and three levels were considered for the last attribute, 'Decision talk' (table 1).

### Willingness-to-pay
There is no previous literature on WTP regarding SDM in BCSP. Therefore, the options were determined according to the average value (€40) of a consultation with a primary care doctor in Spain,[24] and a variation (€30) was considered for the fraction of time devoted to SDM. Moreover, we considered other research papers that have analysed the way to introduce SDM, which indicates that the ideal context is the primary care context, with either a physician or a nurse to apply the SDM model.[25]

### Experimental design
In this phase, attributes and levels are combined to create different choice sets to be evaluated by women. The number of choice sets depends on the number of attributes and their levels. Following Reed Johnson *et al*'s suggestions,[26] four-choice sets were removed because they were neither plausible nor consistent combinations with the theory, leaving them a total of eight choice sets. Those would be eliminated if, without prior discussion, the professional and the woman jointly decided whether to participate or not, no matter if they had received the relevant information via a leaflet or by a healthcare professional.

The eight choice sets presented in pairs resulted in a maximum of 28 tasks (the action of choosing between two alternatives). Many of these could be eliminated, such as in the 16 cases in which one alternative was dominant or dominated by the other. Since SDM is considered a better option to resolve uncertainty in this context,[27] it was considered a dominant choice set. The remaining 12 choice tasks were incorporated into the DCE survey, as this number is below the maximum number of tasks that can be included without causing fatigue on the participants.[28 29]

### Patient and public involvement: piloting and refinement of the questionnaire
In March 2021, a snowball sampling was carried out. Eleven women completed the questionnaire, which helped to clarify its difficulty level and ensure that the questions could be understood. This pre-pilot questionnaire allowed us to identify the difficulties in understanding the elements to be evaluated. The choice tasks were reconfigured from the conventional (structured) arrangement to a descriptive one that narrated a hypothetical scenario with the three attributes in question. To better clarify structured versus descriptive surveys, the two versions are shown in figure 1. Subsequently, a randomised pilot study was conducted on 10 women to evaluate the recruitment process and the survey was performed in a mixed procedure: online (self-applied) and on the telephone (guided).[29]

The survey was conducted between June and November of the same year in both Spanish and Catalan languages. Women were invited to participate, voluntarily and unpaid, through a phone call in which they were informed of the objectives of the research project and if they agreed to participate, then they were asked to complete a survey on the telephone or online. Those who did not complete the survey within 1 week were reminded to do so (figure 2). The 12-choice task corresponding to the DCE included seven questions on socio-demographic aspects (age, nationality, marital status, education, employment status, family history of BC and participation in screening) and also a single, multiple-choice question on WTP for BC screening care with SDM characteristics (online supplemental annex survey).

### Participants
The study was conducted in Catalonia, Spain, with a random sample of women aged between 50 and 60 who

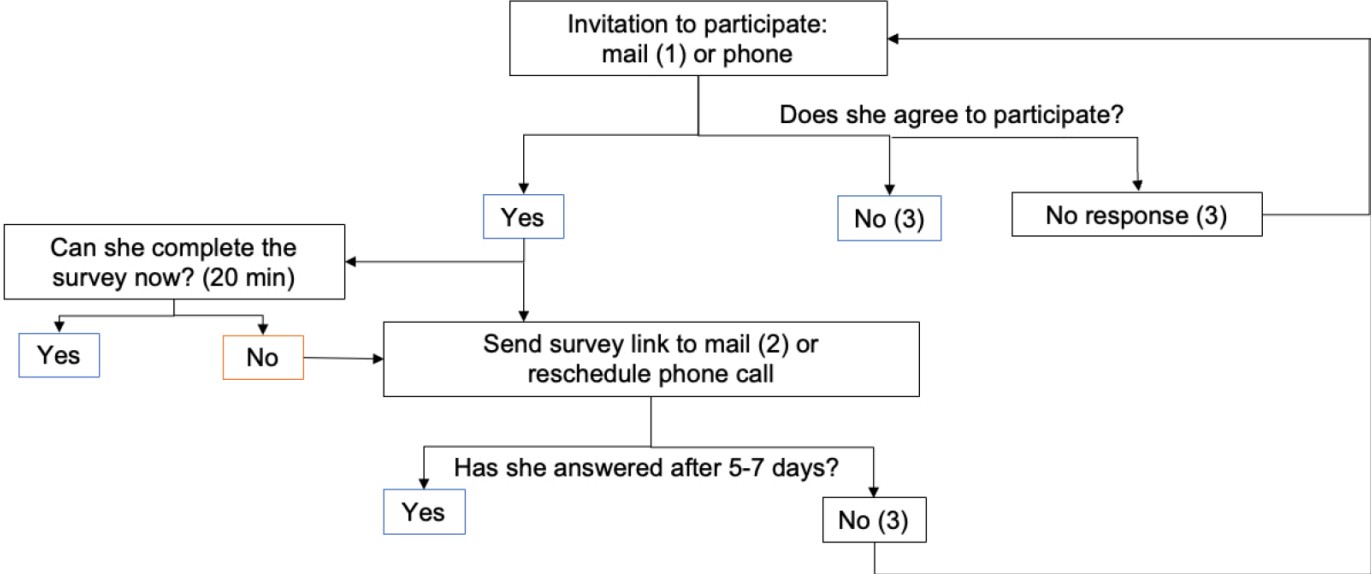

**Figure 2** Net follow-up of survey responses. 1) Attach ethics committe approval. 2) Spanish and Catalan language survey link. 3) If the participants do not respond to the invitation or survey after three attempts, delete them from database.

participated in three BCSPs: at Hospital del Mar in Barcelona, in the Cancer Prevention and Control Program of the Catalan Institute of Oncology and in the Health Region of Lleida. This population had already been invited, at least once, through the national programme to undergo a mammography. Therefore, it was assumed that the topic was more sensitive to their interests. The women who had been diagnosed with BC were excluded from the study.

The minimum required sample size was estimated to be 63 participants, whose calculation was based on the empirical rule recommended by Johnson and Orme.[30]

$N > (500c)/(t \times a)$.

N=number of respondents, t=number of choice tasks, a=number of alternatives and c=maximum number of levels.

### Data analysis

A mixed-effect-conditional logit model was used to estimate women's preferences for different levels of SDM attributes in BC screening. This was based on Daniel McFadden's theory of Discrete Choice, which seeks to describe the behaviour of decision-makers when facing a decision problem, assuming that the declared preferences of those elections are based on the maximum possible-utility achievement.[31] The mixed-effect logistic regression model enables the heterogeneity of preferences in the sample by treating the coefficients as random. It also allows multiple observations from each respondent, which is appropriate for our study, presenting each woman 12 choice tasks. All models included main effects without interaction terms. A model for each subgroup was fitted to allow comparisons according to the WTP.

All attribute variables were coded as dummy variables, with reference levels identified with their results as shown in tables. Furthermore, they were specified as having a

random component, assuming a normal distribution for all model coefficients. These coefficients indicate a change of preference from the reference level for each attribute.[32]

When interpreting the model results, the statistical significance of the coefficients indicates whether attribute levels influence the choice set; whereas the coefficient size indicates the relative importance from one attribute level to another one. We did not include an alternative-specific constant variable because our choice sets were unlabelled. Therefore, they had no utility beyond the attributes assigned to them in the experiment.

For all analyses, statistical significance was set at 0.05. All statistical analyses were performed using the R statistical software.

### RESULTS

Of the 292 women invited to participate, 93 accepted; 27 did not reply, and 66 submitted the survey (reflecting a 70.97% response rate); and 1 being excluded because of a history of BC. Therefore, a total sample of 65 women was obtained; the responses from 2 were collected on the phone, and those from the remaining 63 were collected

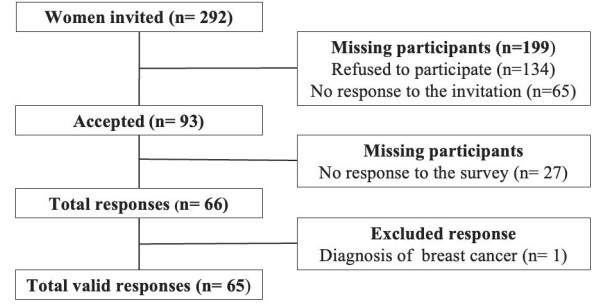

**Figure 3** Flowchart of participants.

**Table 2** Characteristics of participants

| Characteristic | | |
|---|---|---|
| Age | 56.9 | (1.34) |
| Birthplace | | |
| *Spain* | 57 | (87.7%) |
| *Other country* | 8 | (12.3%) |
| Relationship status | | |
| *Single* | 17 | (26.2%) |
| *With a partner* | 48 | (73.8%) |
| Education | | |
| *Primary* | 18 | (27.7%) |
| *Secondary* | 26 | (40.0%) |
| *University* | 21 | (32.3%) |
| Main occupation | | |
| *Unemployed* | 6 | (9.23%) |
| *Paid work* | 48 | (73.8%) |
| *Homemaker* | 11 | (16.9%) |
| Family history of breast cancer | | |
| *No* | 50 | (76.9%) |
| *Yes* | 14 | (21.5%) |
| *Unknown* | 1 | (1.54%) |
| Do you have a mammogram regularly? | | |
| *No* | 11 | (16.9%) |
| *Yes* | 54 | (83.1%) |

Categorical variables are summarised using frequency and percentage. Age is summarised using mean and standard deviation (SD).

via online forms (figure 3). The participants' characteristics are presented in table 2.

## DCE results and trade-off

According to the mixed-effect-conditional logit model, the weighting for each attribute was estimated using the responses to the 12 tasks. There were no missing values among the responses.

The results show that the attribute 'Decision talk' was the only important one for women when considering SDM in BC screening. Women preferred to make decisions alone (coefficient=2.879; 95% CI=2.297 to 3.461) or to engage in SDM with healthcare professionals (coefficient=2.375; 95% CI=1.573 to 3.177), in comparison to having a healthcare professional make the decision for them (table 3). The marginal rate of substitution between the two coefficients for the person who made the decision was 1.21. This meant that women perceive 21% more utility when making decisions than making them with a healthcare professional.

The attributes 'Information talk' and 'Dialogue talk' showed no significant results. Therefore, women do not mind either receiving information on the benefits and risks of mammography through a leaflet or being informed by a healthcare professional (coefficient=−0.168; 95% CI=−0.665 to 0.329). Neither did they show any clear preference for an appointment with a healthcare professional to discuss their beliefs in scheduling their next mammography over a standard screening schedule (coefficient=0.145; 95% CI=−0.373 to 0.663).

## Willingness-to-pay for SDM

More than half of the participants (52.3%) were willing to pay for the SDM process in BC screening; most (38.5%)

**Table 3** Results from the mixed-effects conditional logit model

| Attribute | Levels | Coefficient/SD | SE | P value |
|---|---|---|---|---|
| 1. Information talk | Leaflet | Reference category | – | – |
| | Healthcare professional, mean | −0.168 | 0.2536 | 0.509 |
| | Healthcare professional, SD | 1.473 | 0.3206 | <0.001 |
| 2. Dialogue talk | No, health system schedules mammography | Reference category | – | – |
| | Yes, women share and discuss their beliefs and values about mammography with a healthcare professional, mean | 0.145 | 0.2645 | 0.583 |
| | Yes, women share and discuss their beliefs and values about mammography with a healthcare professional, SD | 1.371 | 0.3212 | <0.001 |
| 3. Decision talk | The healthcare professional makes the decision | Reference category | – | – |
| | Shared decision-making by the healthcare professional and the woman, mean | 2.375 | 0.4093 | <0.001 |
| | Shared decision-making by the healthcare professional and the woman, SD | 1.415 | 0.5208 | 0.007 |
| | The woman makes the decision, mean | 2.879 | 0.2967 | <0.001 |
| | The woman makes the decision, SD | 1.988 | 0.2626 | <0.001 |

SD, standard deviation (of the random effects); SE, standard error.

**Table 4** Women's willingness to pay for shared decision-making in breast cancer screening

| Copayment (euros) | n | % |
|---|---|---|
| 0 | 31 | 47.70 |
| 10 | 6 | 9.23 |
| 20 | 6 | 9.23 |
| 30 | 13 | 20.00 |
| 40 | 2 | 3.08 |
| 60 or more | 7 | 10.80 |

would be willing to pay between €10 and €30, and the remaining ones (13.8%) would be willing to pay €40 (table 4).

Two regression models were fitted for the sample of women according to their WTP: those who showed WTP and those who did not (table 5). The results for both subgroups were similar to those obtained for the entire sample. Thus, the attribute 'who makes the decision' was the only important one for women regardless of their WTP.

In addition, the results for the subgroups show the same trends in women's preferences regarding both 'Information talk' and 'Dialogue talk', as in the regression model for the entire sample.

## DISCUSSION AND CONCLUSION
### Main findings
This study analysed women's preferences for the attributes of SDM—Information talk, Dialogue talk and Decision talk—in a BCSP through a DCE. The main result is that women prefer to make decisions themselves or together with the healthcare professional, while they do not like the healthcare professional making the decision for them. Women perceive more utility in deciding for themselves than in deciding jointly with the healthcare professional. More than half of the women were willing to pay for screening with SDM characteristics. There was no difference in the responses obtained between women with a positive WTP and those who were not willing to pay. Regarding methodology, this study has a strong response rate (>70%) due to the close monitoring of participants' responses.[33]

This study is important for its contribution to the scarce existing literature on the economic quantification of SDM, research which has always been carried out in the form of qualitative studies. Moreover, most studies have focused on its applicability to public health systems; few have examined the topic from an economic perspective, and even fewer have focused on WTP using a DCE methodology.

### Attribute implications
For the first attribute, *Information talk*, the women in our study did not show a clear preference to be given an informational leaflet explaining, in a simple and balanced manner, the benefits and risks of screening instead of having this information explained by a health professional. The lack of difference between the two alternatives may be related to the fact that women do not perceive any difference in the information received. This is supported by studies such as the one conducted by Longo *et al*,[20] who performed a DCE to determine the SDM preferences of patients in the context of chronic diseases. In their study they reported that the second most important attribute for patients was that information could be easily understood.

**Table 5** Results from the mixed-effect-conditional logit model stratified by willingness to pay

| Attribute | Levels | Not willing to pay | | | Willing to pay | | |
|---|---|---|---|---|---|---|---|
| | | Coef./SD | SE | P value | Coef./SD | SE | P value |
| 1. Information talk | Leaflet | Ref. | – | – | Ref. | – | – |
| | Healthcare professional, mean | −0.323 | 0.3706 | 0.383 | −0.074 | 0.3768 | 0.845 |
| | Healthcare professional, SD | 1.537 | 0.4233 | <0.001 | 1.434 | 0.4684 | 0.002 |
| 2. Dialogue talk | No, health system schedules mammography | Ref. | – | – | Ref. | – | – |
| | Yes, women share and discuss their beliefs and values about mammography with a healthcare professional, mean | 0.187 | 0.3944 | 0.636 | 0.136 | 0.3825 | 0.721 |
| | Yes, women share and discuss their beliefs and values about mammography with a healthcare professional, SD | 1.599 | 0.4653 | <0.001 | 1.479 | 0.4756 | 0.002 |
| 3. Decision talk | The healthcare professional makes the decision | Ref. | – | – | Ref. | – | – |
| | Shared decision-making by the healthcare professional and the woman, mean | 2.22 | 0.5852 | <0.001 | 2.826 | 0.6392 | <0.001 |
| | Shared decision-making by the healthcare professional and the woman, SD | 1.449 | 0.56 | 0.01 | 1.649 | 0.8826 | 0.062 |
| | The woman makes the decision, mean | 2.784 | 0.422 | <0.001 | 3.006 | 0.4328 | <0.001 |
| | The woman makes the decision, SD | 1.803 | 0.3734 | <0.001 | 2.548 | 0.441 | <0.001 |

Coef, coefficient; Ref, reference category; SD, standard deviation (of random effects); SE, standard error.

Therefore, using a patient decision aid (PtDA) may help women to better understand the information.

PtDAs deliver balanced information on the risks and benefits of BC screening, which increases IDM.[34] In a recent study in Spain,[35] 23% of the women who received information in this way made informed decisions about their participation in screening, compared with 0.5% in the control group, who had only received information attached to the invitation letter provided by the BCSP. PtDAs have the advantage of being repeatedly consulted by women to review information, as opposed to a clinical appointment. However, the two strategies could act in a complementary fashion, and PtDAs could be used to prepare for the latter appointments to resolve doubts and review the information they contain.[36]

Some institutions, such as the Public BCSP in Catalonia, have already incorporated an invitation letter for screening with a leaflet including information on the explicit need to make a decision, such as overdiagnosis, the balance of benefits and risks and scientific uncertainties.[37] Documentation is scarce in Spain, and until recently, various programmes have not provided balanced information on the benefits and adverse effects of early detection testing.[38]

Improving the information provided in the programme is proposed as a strategy of PtDAs, which can be used with individuals of different literacy levels; in the local context of this study, they would ideally be used with women of a higher educational level because they are really interested in an SDM,[20] and are able to understand complex information such as prevalence and absolute risks[39 40] without needing an explanation by a healthcare professional.

Regarding the second attribute, *Dialogue talk*, the women in our study were indifferent in regard to the choice between attending a face-to-face appointment with a healthcare professional to discuss their beliefs and concerns about participation in the screening and the standard approach, in which the healthcare system schedules the screening. These results may be due to the wide acceptance of screening among women not only because of the well-known benefits but also the minimisation of risks; information which has been disseminated by persuasive preventive public health campaigns of different governments. This has led women to unquestionably participate in screening and more openly discuss the benefits and risks as well as and on their beliefs and fears.[41]

However, in a study by Longo *et al*,[2020] the most relevant attribute for patients was their need to feel heard by their physicians, particularly because patients reported the feeling of not being listened to.[42] Thus, communication remains an element that should be considered by healthcare professionals when they interact with their patients. Showing a receptive attitude, acceptance, empathy and lack of judgement are the first steps in generating a trustful relationship and thus advancing to a more participatory health model.[43] Without this, it is impossible to have awareness of the beliefs of patients and, therefore,

encourage them to participate in the process of SDM. This requires training in competences and communication skills.[42]

Conversely, there is a high probability that women already have beliefs or fears regarding BC and want to discuss them with a health professional. In particular, the fact that having BC and having survived it is not unusual,[6] it means that women usually know other women already diagnosed with or treated for BC. In our study, one out of five women had a first-degree relative with a history of BC. This would increase, if the scope were extended to other affective bonds such as friends and acquaintances. This is interesting when considering personalisation of the screening, in which women must be assisted by a healthcare professional to determine whether their risk is high, medium or low, and thus recommendations must be made for the frequency of screening through an SDM process.[44]

Regarding the third attribute, *Decision talk*, women reject paternalistic health models, in which the healthcare professional or health system exclusively makes health decisions. This result is consistent with the current trend of empowering patients to be more autonomous in clinical decisions.[34 37] However, this degree of autonomy may be unrealistic because of the lack of knowledge regarding adverse effects.[45] For example, a recent study indicated that Spanish women have low awareness of overdiagnosis.[46 47] Only 8.1% knew the meaning of overdiagnosis, even though, the percentage increased to 54.2% in women who had received PtDAs.[47]

In Longo *et al*'s study,[20] patients preferred to make decisions for themselves or with a healthcare professional and rejected the approach of the decision being made exclusively by a professional, which accords with our results. However, in Longo's study, this attribute was one of the least important.

The characteristics of the clinical process differ either when deciding on treatment or when working in an early detection context, in which the target population is healthy. An option well-evaluated by women in previous studies[48] is recommending the development of a targeted SDM to specific groups of women, such as those who need more information on the benefits and risks, those who have their first experience with screening, those who have lower educational levels and those who have a high risk of being diagnosed with BC. All of them require face-to-face appointments with healthcare professionals.[36 49] However, the implementation of IDM for women with a high educational level and a low risk of being diagnosed with BC would be sufficient, and an appointment with a healthcare professional would be unnecessary.[50] This second group can be given PtDAs with simple and clear information on screening for them to make the decision autonomously,[34 51 52] which implies a lack of follow-up by the healthcare professional. A study is currently being conducted at the Mayo Clinic to determine whether women's discussion groups for decision-making could provide a new line to support and prepare women for

deciding on BC screening.[53] This is based on the strategy mentioned above: the PtDAs could be delivered before the clinical meeting, and subsequently there would be a conversation with other women about the information provided there. As such, the appointment between the healthcare professional and the woman would only aim at clearing up doubts or concerns originated from the PtDAs.[36]

In conclusion, the results of the DCE allow us to analyse SDM barriers and the difficulties in its applicability[40 54 55] and because of this, the main approaches have been developed within contexts in which decision-making is particularly difficult.[55] Besides, there are still few examples of studies like ours to be applied in primary care, where patients make less difficult decisions, yet, not less important for their impact on people's quality of life.[51]

### Willingness to pay

The subgroups, separating those who were or were not willing to pay, showed a preference for making decisions jointly or alone. No studies were found regarding WTP for SDM in the context of BC screening. Only one article for patients with BC diagnosis was found, but the results focused on WTP according to the type of treatment and not on SDM characteristics.[56] Another study, focused on prostate cancer, reported that participants with high WTP preferred an active rather than a passive participation model.[57]

Regarding the monetary value of SDM, this study's findings can be compared with the results of Wilson *et al*'s[57] study on patients with prostate cancer in the USA. It was determined that men had a WTP between US$25 and US$50 for the implementation of SDM.[57] WTP was associated with marital status (single people valued SDM more highly), the stage of SDM (those who were initiating the process of decision-making were more likely to be willing to pay compared with those who were close to deciding) and the participation model (more active patients are more likely to be willing to pay compared with passive ones).[57]

In addition, Brito *et al*[58] evaluated patient-reported experiences among the adult population in outpatient care through a DCE, in which SDM characteristics were included. In this case, respondents were willing to pay, on average, €16 for a doctor who provided easy-to-understand explanations in comparison to one who did not provide understandable ones. They were also willing to pay, on average, €20 for physicians to whom they could ask questions or raise concerns as opposed to physicians who focused on providing information. Finally, they were willing to pay, on average, €22 to physicians who involved the patient in making decisions, in contrast to those who made decisions on their own.[58]

Finally, it should be considered that in Spain, screening is fully covered by the National Health System (NHS) for women aged between 50 and 69. Although it is free of cost, the ability to pay does not effectively coincide with the real payment, WTP shows how much society is willing

to pay for a healthcare innovation; specifically, in our case, its WTP for a relationship between professional and patient in the clinical appointments.[59] Moreover, in the case of innovation being implemented in the NHS, it may imply a burden on the health system and copayment or direct payment from women.[14] In the last case, the out-of-pocket cost for a screening is one of the determining factors for adherence to SDM in BCSP.[60]

Future research could include randomised clinical trials, in which women would experience healthcare with all the attributes that SDM elicits; their preferences, perceived satisfaction and health benefits could be analysed in comparison to regular attendance; and the costs that the implementation of SDM would entail in the screening of BC for the NHS could be identified.

### Limitations

The main limitation of the study, which forced us to change the design of the questions, was the difficulty women experienced when comparing hypothetical profiles. We detected difficulties in understanding the differences between the profiles proposed in each choice task. In addition, their ability to detect differences among the 12 questions in the DCE survey was limited. This may also be due to women's lack of familiarity with participatory health characteristics because of the current BCSP model (exposure bias). Moreover, the decisions regarding screening seem to reflect a social consensus on the need for screening and thus its utility is not questioned. To eliminate exposure bias, future randomised studies should be conducted among women who can experience both models of care (paternalistic and SDM), and then, their preferences and perceived satisfaction can be compared with their experience of regular attendance to identify the costs that the implementation of SDM would entail in the BCSP. Considering that 'who makes the decision' is the most important factor for women, strategies could be proposed to ensure that through discussion groups, women can make an informed decision, as contexts of prevention, in contrast to those of treatment, which may be more accepted by women.[53] These new procedures are currently being introduced in studies conducted at the Mayo Clinic, among others which aimed at facilitating more participatory and more informed decision-making.[53]

Finally, it is difficult to precisely determine the ability to pay because in Spain, BC screening is within the purview of the NHS. Therefore, services are free of cost at the time they are provided; and the population assumes that this is a consolidated health benefit and consequently something not to be paid for, in particular because screening for other types of cancer (eg, colorectal) is currently being introduced and heavily promoted by the NHS.

### Conclusion

This is the first study examining women's preferences regarding SDM in relation to BC screening. Women were shown to reject paternalistic health models in the context of BCSP and to prefer models with IDM or SDM. More

than half of the participants were willing to pay for active involvement in their health decisions.

## Practical implications

The change from a paternalistic model to a participatory model of person-centred medicine would require restructuring the BCSP so that women can make informed decisions for themselves or engage in SDM together with a healthcare professional.

**Author affiliations**
¹Department of Economics, University Rovira i Virgili, Reus, Spain
²Research Centre on Economics and Sustainability (ECO-SOS), Reus, Spain
³Research Group on Statistics, Economic Evaluation and Health (GRAEES), Reus, Spain
⁴Area Q: Evaluation and Research in the Field of Social Sciences and Health, Barcelona, Spain
⁵School of Nursing and Occupational Therapy (EUIT), Autonomous University of Barcelona, Terrasa, Spain
⁶Cancer Prevention and Control Program, Catalan Institute of Oncology-IDIBELL, L'Hospitalet de Llobregat, Spain
⁷Department of Basic Medical Sciences, University of Lleida-IRB, Lleida, Spain
⁸Department of Basic Medical Sciences, University of Lleida, Lleida, Spain
⁹HEOR freelance consultant, Heorfy Consulting, Reus, Spain

**Contributors**  MJH-L (guarantor), MJP-L and MC-L: Conceptualisation, Methodology, Formal analysis, Investigation, Project administration, Supervision, Visualisation, Writing—original draft. AC and NC-B: Data curation, Formal analysis, Writing—review and editing. CV: Resources, Writing—review and editing. MR: Resources, Formal analysis, Writing—review and editing. CF: Software, Formal analysis, Writing—review and editing.

**Funding**  Financial support for this study was entirely provided by a grant from Instituto de Salud Carlos III through the project PI18/00773 (co-funded by European Regional Development Fund) and the European Union's Horizon 2020 research and innovation programme under Marie Skłodowska-Curie grant agreement No. 713679 from the Universitat Rovira i Virgili (URV). The funding agreement ensures the authors' independence in designing the study, interpreting data, and writing and publishing the report.

**Competing interests**  The authors declare that they have no conflict of interest.

**Ethics approval**  This research was approved by the Medicinal Product Research Ethics Committee (CEIm) of the Institut d'Investigació Sanitària Pere Virgili (Pere Virgili Health Research Institute) Ref. CEIM: 175/2019. Participants in the informed consent agreed to participate in the study and it was established that the responses were confidential and would be used only for this research. Informed consent in the case of telephone participants was verbal. While those who did it online had to accept to be part of the study through the virtual platform previous to their accessing the survey.

**Provenance and peer review**  Not commissioned; externally peer reviewed.

**Data availability statement**  All data relevant to the study are included in the article or uploaded as supplementary information.

**ORCID iDs**
María José Hernández-Leal http://orcid.org/0000-0002-4002-6454
María José Pérez-Lacasta http://orcid.org/0000-0001-5906-5632
Núria Codern-Bové http://orcid.org/0000-0003-0210-7488
Carmen Vidal-Lancis http://orcid.org/0000-0003-2768-2710
Montserrat Rue http://orcid.org/0000-0002-7862-9365
Carles Forné http://orcid.org/0000-0002-8133-3274
Misericòrdia Carles-Lavila http://orcid.org/0000-0003-3796-3014

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
