## [Reviewer comments · BMJ Open]

ARTICLE DETAILS

TITLE (PROVISIONAL)	Women's Preference to Apply Shared Decision Making in Breast Cancer Screening: A Discrete Choice Experiment
AUTHORS	Hernández Leal, María José; Pérez-Lacasta, María José; Cardona, Angels; Codern-Bové, Núria; Vidal, Carmen; Rue, Montserrat; Forné, Carles; Carles-Lavila, Misericòrdia

VERSION 1 – REVIEW

REVIEWER	Maes-Carballo, Marta University of Granada
REVIEW RETURNED	16-Jun-2022

GENERAL COMMENTS	Thank you for inviting me to review this manuscript and I want to congratulate the authors for such an interesting manuscript. Shared decision making (SDM) is a crucial component of evidence-based, a cornerstone for ensuring high-quality cancer care and patient-centred care. Nowadays, its application in cancer care and screening is still scarce, and it faces many difficulties and barriers to overcome. I do not think this manuscript is ready for publication as it suffers major problems. I encourage authors to improve their work following the next instructions. I highlighted some of them: In general, the study is very specific. It is focused in Catalonia and its screening program and it is only useful for that public. I think they should open the study to an international public or at least specified the type of screening Catalonian Health care offer. I reckon it should be required more countries and areas participating in this study for improving external validity. General: - English writing must be improved. Abstract: - Results should show some quantities, percentages. It everything very generic. - Abstract should be rewritten. Results and conclusions should be improved. Introduction: - I will suggest adding what kind of screening is performed in Catalonia or in the countries the study will be focused. Methods: - It is well-developed.
--

	- Data collection page 5: Could you show us the semi-structured interview? Please add it as Appendix. - There is a possibility of major bias as participants have received money for participation. Results: - “Willingness to pay for SDM: More than half of the participants (52.3%) were willing to pay for the SDM process in BC screening; most (38.5%) would be willing to pay between 10€ and 30€, and the rest (13.8%) would be willing to pay 40€. (Table 4)” How did you choose the quantities? Why did you choose these quantities of money and not others? Why do you think it is important the differences between 10 € less or more? Is that really relevant? Discussion - “Of the 292 women invited to participate, 134 refused, 92 accepted but did not reply, and 66 submitted the survey (reflecting a 22.6% response rate)”. Why the participation rate is too low? Is that a limitation of the study? How the authors work against it? - “Women perceive 20 percent more efficacy in deciding than in making it jointly with the health professional”. Main findings must not have quantities, ... - What are the strengths of the study? - Limitations:  o “The main limitation of the study was the difficulty for women to compare hypothetical profiles”. How did you solve this limitation? o Authors did not state any inconvenience about the low rate of response. o Limitations should improve writing and must try to give solutions. It is one of the most important parts of the manuscript to understand the risk of bias of this publication. Conclusions: - “This is the first study to determine women’s stated preferences for BC screening when SDM is applied. Women reject a paternalistic health model and favour one with SDM or a variation of it (IDM)”. Is this true? Your work stated that “they prefer to make the decision by themselves -applying an Informed Decision making”. Informative doctor patient relationship is not a variation of SDM. It is a different kind of relationship between doctor and patient. - It should be rewritten. It is too long and must focused on the findings and why this work is important for research. - “Future research could include randomised clinical trials, in which women would experience health care with all the attributes that SDM elicits and then analyse their preferences, perceived satisfaction, and profit in contrast to regular attendance, and to identify the costs that the implementation of SDM would entail in the screening of BC for the NHS”. This statement should be placed in the discussion. In the Conclusion section should be less extent. - DATA SHARING STATEMENT  o “All data relevant to the study are included in the manuscript and tables or uploaded as supplementary information”. Is this true? I reckon some information might be missed.
--	--

REVIEWER	Hersch, Jolyn University of Sydney, School of Public Health
-----------------	--

GENERAL COMMENTS

This manuscript presents a discrete choice experiment investigating women's preferences regarding a shared decision making approach to considering participation in breast cancer screening, in comparison with the current approach used in the Catalonian breast screening program in Spain. The research question is an interesting one; I am not aware of any previous literature addressing it.

While I fully acknowledge the additional challenge faced by the authors in having to write in a non-native language, unfortunately in many instances the awkward written expression makes it difficult to understand the manuscript. Therefore, this aspect requires some close attention, probably with the help of a native speaker, in order to prepare the manuscript (including all components) for thorough peer review and hopefully future publication. Below are some specific comments.

1. The Abstract is quite difficult to understand, at least for somebody like myself who does not have very detailed familiarity with DCEs. I also noticed it is quite short and not structured according to the BMJ Open author instructions. Please revise this very important part of the manuscript to make the results and conclusions more understandable.
2. Similarly, the Strengths and Limitations bullet points are supposed to be written relating specifically to the methods – not the results – of the study.
3. Background – page 4 line 31. “agreement...to undergo mammography” should refer to an agreement/decision to either undergo it or not to undergo it.
4. Background – Somewhere please explain just a little (perhaps one sentence) what you mean by “usual care with characteristics of a paternalist model” in this context (i.e. how paternalism contrasts with SDM).
5. Section 2.1 – I found this whole section quite confusing until I had read it over a few times. Perhaps it could be made clearer by expanding Table 1 to contain more text but showing how each level of each attribute fits in with either the current screening program or the SDM model? The appendix is helpful to understand the task for participants.
6. Section 2.1 – page 5 line 34. I think “decide to participate” should be “decide whether or not to participate”.
7. Section 2.3 – Please edit this to make the meaning clearer: “the previous formulation of the questions had a conventional structured form to a descriptive form that included all the three attributes in one sentence”.
8. Please add a sentence or two explaining how and when the potential participants were approached to invite them to take part.
9. Table 2 – Remove the characteristic Paid Work which is redundant with Main Occupation.

	10. Table 3 and Table 5 – Again I acknowledge I am not an expert in DCEs (and the associated statistics), but is it necessary to include the ‘SD’ row as well as the ‘mean’ row for each item? The ‘mean’ numbers seem to be the only ones referred to and interpreted in the text. What is the importance/relevance of the ‘SD’ numbers? 11. Section 4.1 – page 13 line 5. “Women perceive 20 percent more efficacy” doesn’t really make sense... I think the authors mean utility rather than efficacy, but I’m not sure whether the finding can be stated in some other way to make its meaning and interpretation clearer. 12. Section 4.2 – page 13 line 38. In mentioning the NHS, please state which country this refers to. Section 4.2 – page 13 lines 46-54. To me, this paragraph seems to come out of nowhere... It starts “Conversely...” but I don’t understand how it relates to the preceding text. 13. Page 14 line 4 – “presential” – Is this a real word? And line 23 – what are “transversal skills”? 14. Limitations – The authors acknowledge an important limitation resulting from participants’ difficulties in understanding the differences between the profiles etc. It would be good to suggest what effect this may have had on the findings (e.g. biased towards the null because it limited the capacity to identify meaningful preferences regarding the other attributes). 15. Conclusion – “stated preferences for BC screening when SDM is applied” is not quite right; the study concerns preferences about SDM regarding screening (not preferences about screening).
--	---

VERSION 1 – AUTHOR RESPONSE

Reviewer: 1
Dr. Marta Maes-Carballo, University of Granada

Reviewer comments	Authors responds
General: 1. English writing must be improved.	The article was initially edited in English by the ELSEVIER language service. We solicited a new correction.
Abstract: 2. Results should show some quantities, percentages. It everything very generic. 3. Abstract should be rewritten. Results and conclusions should be improved.	2. The section was rewritten: Objective: To analyse women’s stated preferences for establishing the relative importance of each attribute of Shared Decision making (SDM) and their willingness to pay (WTP) for more participatory care in a breast cancer screening programme (BCSP).

	Design: A Discrete Choice Experiment was designed with 12 questions (choice tasks). It included three attributes: 'How the information is obtained' regarding benefits and harms, whether there is a 'Dialogue for mammography scheduled' between the healthcare professional and the woman, and, 'Who makes the decision' regarding participation in BCSP. The data were obtained using a survey that included 12 choice tasks, one question on WTP, and seven socioeconomic questions. The analysis was performed using conditional mixed-effects logit regression and stratification according to WTP. Setting: Data collection related to BCSP was conducted between June and November 2021 in Catalonia, Spain. Participants: Sixty-five women between 50 and 60 years old. Main outcome measures: Women's perceived utility of each attribute, trade-off on these attributes, and WTP for SDM in BCSP. Result: The only significant attribute was 'Who makes the decision'. The decision made by oneself alone (coefficient=2.879; 95% CI=2.297, 3.461) and the decision made together with a healthcare professional (2.375; 95%CI=1.573, 3.177) were the options preferred by women. The former contributes 21% more utility than the latter. Moreover, 52.3% of women stated a WTP 10 or more euros for SDM. Women's preferences regarding attributes did not influence their WTP. Conclusions: The women refused a current paternalistic model and prefer SDM or Informed Decision making for BCSP.
Introduction: 4. I will suggest adding what kind of screening is performed in Catalonia or in the countries the study will be focused.	4. The description of breast cancer screening in the region of Catalonia in Spain was already included: The screening programme in the Spanish region of Catalonia is organised by the public health system. This screening programme entails sending a postal letter to women between 50 and 69 years of age every 2 years to undergo mammography in a health centre free of charge (4). Several research and communication campaigns have focused on increasing women's adherence to screening to raise awareness of its importance, which led to the Spanish coverage reaching 81.5% in 2017 in the target population (although it had dropped to 73.7% 3 years later, during the pandemic) (5). The most cited reason for attending is the invitation letter issued by the organisations affiliated with the screening programme (6).

	There is a dearth of studies focusing on awareness of patients' preferences in SDM (20), and there are none on women's preferences regarding breast cancer screening programmes (BCSP) and their willingness to pay (WTP) for them. Therefore, this study aimed to analyse the declared preferences of women in Catalonia regarding the attributes of more participatory care (SDM) in contrast to the usual care, which reflects characteristics of a paternalist model for BC screening, by establishing the relative importance of each through a DCE. In addition, the study participants were asked about their WTP for this type of healthcare. In addition, in the section 2.1. in methodology we have included a Table 1 more information where the current model vs the SDM is characterized.
Methods: 5. It is well-developed. 6. Data collection page 5: Could you show us the semi-structured interview? Please add it as Appendix. 7. There is a possibility of major bias as participants have received money for participation.	5. Thank you for your comment 6. It is not possible to add an appendix to the semi-structured interview requested since only quantitative methods (survey) were used, but we have included as an appendix the survey sent to women. 7. There was no financial reward, but this has been mentioned in a section of the article: The women were invited to participate, voluntarily and unpaid, through a telephone call during which they were informed of the objectives of the research project and were asked to complete a survey via telephone or online if they agreed to participate. Those who did not complete the survey within one week were reminded (Figure 2).
Results: -8. "Willingness to pay for SDM: More than half of the participants (52.3%) were willing to pay for the SDM process in BC screening; most (38.5%) would be willing to pay between 10€ and 30€, and the rest (13.8%) would be willing to pay 40€. (Table 4)" How did you choose the quantities? Why did you choose these quantities of money and not others? Why do you think it is important the differences between 10 € less or more? Is that really relevant?	8. There is no literature on the WTP regarding the TOC for BCSP, we thought of using as a reference the value of a clinical consultation in Primary Care (is 40 euros). It has been considered that in other work has worked on how to introduce the SDM, it seems that the ideal context would be in a context of Primary Care, being a doctor or nurse. The following phrase was added in the new section 2.2 Willing-to-pay of the methodology: There is no previous literature on WTP regarding SDM in BCSP. Therefore, the options were determined according to the average value (40€) of a consultation with a primary care doctor in Spain (24), and a variation (30€) was

	considered for the fraction of time devoted to SDM. Moreover, we considered other works that have analysed how to introduce SDM, which indicate that the ideal context is the primary care context with either a physician or a nurse for applicate the SDM model (25).
Discussion 9. "Of the 292 women invited to participate, 134 refused, 92 accepted but did not reply, and 66 submitted the survey (reflecting a 22.6% response rate)". Why the participation rate is too low? Is that a limitation of the study? How the authors work against it? 10. "Women perceive 20 percent more efficacy in deciding than in making it jointly with the health professional". Main findings must not have quantities, ... 11. What are the strengths of the study?	In the first draft, we had miscalculated the ratio. This is because the total number of women invited to participate was considered, which is a mistake, since women who were invited to participate also received the survey should be considered. As confirmed in the article Watson, Verity; Becker, Frauke; de Bekker-Grob, Esther (2016). Discrete Choice Experiment Response Rates: A Meta-analysis. Health Economics, (), -. doi:10.1002/hec.3354 According to, it was calculated on a base of 93 women, with the real ratio of 70.97% The correction was made in the article: Of the 292 women invited to participate, 93 accepted; 27 did not reply, and 66 submitted the survey (reflecting a 70.97% response rate), with one excluded due to a history of BC. 10. For utility studies (DCE method) one of the main and most relevant results in the trade-off and determine what are the most valued characteristics by users, and to value it is necessary the MRS, which is calculated in relation to the coefficient of the attributes. 11. The following paragraph was included in section 4.1 Main findings. The importance of this study is its contribution to the scarce existing literature on the economic quantification of SDM, research on which has always been carried out in the form of qualitative studies. Moreover, most studies have focused on its applicability to public health systems; few have examined the topic from an economic perspective, and even fewer have focused on WTP using a DCE methodology.
Limitations: 12. "The main limitation of the study was the difficulty for women to compare hypothetical profiles". How did you solve this limitation? 13. Authors did not state any inconvenience about the low rate of response.	12. As described in the section 2.3 Pre-test and refinement of the questionnaire, at the pre-pilot, the difficulties of understanding the elements to be evaluated by women were detected, for this purpose it is mentioned in the same section that the questionnaire was reconfigured from a conventional (structured) to a

14. Limitations should improve writing and must try to give solutions. It is one of the most important parts of the manuscript to understand the risk of bias of this publication.	descriptive one that narrates a hypothetical scenario with the three attributes in question. In order to better clarify in this section what is a structured survey vs the descriptive one and the two versions were added in Figure 1. Along with the above mentioned is mentioned in the limitations section: This may also be due to women’s lack of familiarity with participatory health characteristics due to the current BCSP model (exposure bias) as well as that the decisions regarding screening seem to reflect a social consensus on the need for screening and that its utility is not questioned. 13. There is no longer a problem of the response rate with the recalculation 14. No other important biases were found for the study, however a way to avoid exposure bias in the attention model is proposed. To eliminate exposure bias, future randomised studies should be conducted in which women can experience both models of care (paternalist and SDM), and then their preferences and perceived satisfaction can be compared to their experience of regular attendance to identify the costs that the implementation of SDM would entail in the BCSP. Considering that ‘who makes the decision’ is the most important factor for women, strategies could be proposed to ensure that, through discussion groups, women can make an informed decision, as contexts of prevention, in contrast to those of treatment, may be more acceptable to women (53). These new procedures are currently being introduced in studies being conducted at the Mayo Clinic, among others, which are aimed at facilitating more participatory and more informed decision making (53).
Conclusions: 15. “This is the first study to determine women’s stated preferences for BC screening when SDM is applied. Women reject a paternalistic health model and favour one with SDM or a variation of it (IDM)”. Is this true? Your work stated that “they prefer to make the decision by themselves -applying an Informed Decision making”. Informative doctor patient relationship is not a variation of SDM. It is a different kind of relationship between doctor and patient.	15. Thank you very much for the comment, and the correction has been made in the text: Women were shown to reject paternalistic health models in the context of BCSP and to prefer models with IDM or SDM. 16. The paragraph on future studies was incorporated in the Discussion section, thus shortening the conclusion. 17. We have included in the limitations section a sentence on this subject:

16. It should be rewritten. It is too long and must focused on the findings and why this work is important for research. 17. "Future research could include randomised clinical trials, in which women would experience health care with all the attributes that SDM elicits and then analyse their preferences, perceived satisfaction, and profit in contrast to regular attendance, and to identify the costs that the implementation of SDM would entail in the screening of BC for the NHS". This statement should be placed in the discussion. 18. In the Conclusion section should be less extent.	To eliminate exposure bias, future randomised studies should be conducted in which women can experience both models of care (paternalist and SDM), 18. The paragraph on future studies was incorporated in the Discussion section, thus shortening the conclusion.
--	--

Reviewer: 2

Dr. Jolyn Hersch, University of Sydney

Reviewer comments	Authors responds
1. The Abstract is quite difficult to understand, at least for somebody like myself who does not have very detailed familiarity with DCEs. I also noticed it is quite short and not structured according to the BMJ Open author instructions. Please revise this very important part of the manuscript to make the results and conclusions more understandable.	We rewrote the seccion: Objective: To analyse women's stated preferences for establishing the relative importance of each attribute of Shared Decision making (SDM) and their willingness to pay (WTP) for more participatory care in a breast cancer screening programme (BCSP). Design: A Discrete Choice Experiment was designed with 12 questions (choice tasks). It included three attributes: 'How the information is obtained' regarding benefits and harms, whether there is a 'Dialogue for mammography scheduled' between the healthcare professional and the woman, and, 'Who makes the decision' regarding participation in BCSP. The data were obtained using a survey that included 12 choice tasks, one question on WTP, and seven socioeconomic questions. The analysis was performed using conditional mixed-effects logit regression and stratification according to WTP. Setting: Data collection related to BCSP was conducted between June and November 2021 in Catalonia, Spain. Participants: Sixty-five women between 50 and 60 years old.

	Main outcome measures: Women's perceived utility of each attribute, trade-off on these attributes, and WTP for SDM in BCSP. Result: The only significant attribute was 'Who makes the decision'. The decision made by oneself alone (coefficient=2.879; 95% CI=2.297, 3.461) and the decision made together with a healthcare professional (2.375; 95%CI=1.573, 3.177) were the options preferred by women. The former contributes 21% more utility than the latter. Moreover, 52.3% of women stated a WTP 10 or more euros for SDM. Women's preferences regarding attributes did not influence their WTP. Conclusions: The women refused a current paternalistic model and prefer SDM or Informed Decision making for BCSP.
2. Similarly, the Strengths and Limitations bullet points are supposed to be written relating specifically to the methods – not the results – of the study. 3. Background – page 4 line 31. “un...to undergo mammography” should refer to an agreement/decision to either undergo it or not to undergo it.	2. The section was rewritten:  - DCEs have limitations on sample size, and SDM, as a little-known model, makes it difficult for participants to understand the hypothetical situations described in the choice task. - A DCE is an indirect method to obtain stated preferences and information that are otherwise impossible to reveal when actual choice behaviour is restricted in some way. - The follow-up plan of this study based on the response level of the participants allowed us to obtain a strong response rate. - The responses obtained on participants' WTP may be sensitive/not robust when performed in a public health system context. - This is the first study concerning women's preferences related to SDM regarding BCSPs. 3. The section was rewritten: SDM allows women, based on their own beliefs and values, to decide, together with a health

	professional, whether to undergo mammography
4. Background – Somewhere please explain just a little (perhaps one sentence) what you mean by “usual care with characteristics of a paternalist model” in this context (i.e. how paternalism contrasts with SDM).	4. An explanatory column between the two models (SDM and paternalist) was included in Table 1 to clarify the attributes. Also, the section was rewritten: SDM is characterised as a participatory care model, in which the professional and the patient are on par with each other in terms of power to deliberate regarding the best decision. This contrasts with the paternalistic health model, in which the health professional or the system makes such decisions (11). The latter phenomenon remains in force in some health programmes despite declarations and intention to shift to a more participatory one.
5. Section 2.1 – I found this whole section quite confusing until I had read it over a few times. Perhaps it could be made clearer by expanding Table 1 to contain more text but showing how each level of each attribute fits in with either the current screening program or the SDM model? The appendix is helpful to understand the task for participants.	5. An explanatory column between the two models (SDM and paternalist) was included in Table 1 to clarify the attributes. This section was also rewritten: . Table 1 describes the levels of a participatory healthcare model with characteristics of SDM (a health professional informs the patient of the risks and benefits, and a dialogue is held between the woman and the health professional, who jointly make the decision) as well as of the current model of care, which has characteristics of a paternalistic model (the risks and benefits are provided via a leaflet, the healthcare system schedules mammography according to age criteria, and the health professional makes the decision). Moreover, included in the third attribute (‘Decision talk’) is an additional level wherein the woman makes a decision for herself, that is, informed decision-making (IDM).
6. Section 2.1 – page 5 line 34. I think “decide to participate” should be “decide whether or not to participate”.	The section has been redrafted and a definition has been included in Table 1.
7. Section 2.3 – Please edit this to make the meaning clearer: “the previous formulation of the questions had a conventional structured form to a descriptive form that included all the three attributes in one sentence”.	7. En la figura 1 se incluyó un ejemplo de una choice task convencional-estructurada y una narrativa-descriptiva.

	In addition, the following phrase was added to the manuscript: This pre-pilot allowed us to identify the difficulties in understanding the elements to be evaluated. The choice tasks were reconfigured from the conventional (structured) arrangement to a descriptive one that narrated a hypothetical scenario with the three attributes in question. To better clarify structured vs descriptive surveys, the two versions are shown in Figure 1.
8. Please add a sentence or two explaining how and when the potential participants were approached to invite them to take part.	A new Figure 2 presenting the invitation and follow-up of the participants was incorporated. In addition, the following sentence was inserted in section 2.4 : The women were invited to participate, voluntarily and unpaid, through a telephone call during which they were informed of the objectives of the research project and were asked to complete a survey via telephone or online if they agreed to participate. Those who did not complete the survey within one week were reminded (Figure 2).
9. Table 2 – Remove the characteristic Paid Work which is redundant with Main Occupation.	We removed
10. Table 3 and Table 5 – Again I acknowledge I am not an expert in DCEs (and the associated statistics), but is it necessary to include the ‘SD’ row as well as the ‘mean’ row for each item? The ‘mean’ numbers seem to be the only ones referred to and interpreted in the text. What is the importance/relevance of the ‘SD’ numbers?	10. From the statistical point of view, SD is necessary because it allows to know the variability of the individual random effects in relation to extrapolating the results to the population. In addition, this information could be useful for the calculation of the sample size for new DCE research using the same attributes of our study.
11. Section 4.1 – page 13 line 5. “Women perceive 20 percent more efficacy” doesn’t really make sense... I think the authors mean utility rather than efficacy, but I’m not sure whether the finding can be stated in some other way to make its meaning and interpretation clearer.	11. Thank you very much for the clarification, it actually refers to a 20 percent more utility, because the calculation was made through the marginal rate of substitution (MRS). Correction in the manuscript: Women perceive more utility
12. Section 4.2 – page 13 line 38. In mentioning the NHS, please state which country this refers to. Section 4.2 – page 13 lines 46-54. To me, this paragraph seems to come out of nowhere... It starts “Conversely...”	12. Correction in the manuscript Public BCSP in Catalonia Correcció del párrafo:

but I don't understand how it relates to the preceding text.	Improving the information provided in the programme is proposed as a strategy of PtDAs, which can be used with individuals of different literacy levels; in the local context of this study, they would ideally be used with women of higher educational levels because they are the most interested in an SDM (20), and they can understand complex information such as prevalence and absolute risks (39,40) without the need for an explanation from a health professional.
13. Page 14 line 4 – “presential” – Is this a real word? And line 23 – what are “transversal skills”?	13. Correction in the manuscript: face-to-face appointment Clarification in the manuscript: This requires training in competences and communication skills (42).
14. Limitations – The authors acknowledge an important limitation resulting from participants' difficulties in understanding the differences between the profiles etc. It would be good to suggest what effect this may have had on the findings (e.g. biased towards the null because it limited the capacity to identify meaningful preferences regarding the other attributes).	14. Incorporation in the manuscript: This may also be due to women's lack of familiarity with participatory health characteristics due to the current BCSP model (exposure bias) as well as that the decisions regarding screening seem to reflect a social consensus on the need for screening and that its utility is not questioned. To eliminate exposure bias, future randomised studies should be conducted in which women can experience both models of care (paternalist and SDM), and then their preferences and perceived satisfaction can be compared to their experience of regular attendance to identify the costs that the implementation of SDM would entail in the BCSP.
15. Conclusion – “stated preferences for BC screening when SDM is applied” is not quite right; the study concerns preferences about SDM regarding screening (not preferences about screening).	15. Thanks for the appreciation, we made the change. This is the first study examining women's preferences regarding SDM in relation to BC screening

Editor Comments to authors

Comments	Authors responds
----------	------------------

1. Please clarify whether the consent obtained in this study was written or verbal. If verbal, please clarify if this method was approved by the ethics committee.	1. The Ethics Committee allowed us to give oral consent, as no sensitive data or the names of the study participants would be requested.
2. Please revise the 'Strengths and limitations of this study' section of your manuscript (after the abstract). This section should contain up to five short bullet points, no longer than one sentence each, that relate specifically to the methods. The novelty, aims, results or expected impact of the study should not be summarised here.	This section was rewritten in relation to the methods and design:  - DCEs have limitations on sample size, and SDM, as a little-known model, makes it difficult for participants to understand the hypothetical situations described in the choice task. - A DCE is an indirect method to obtain stated preferences and information that are otherwise impossible to reveal when actual choice behaviour is restricted in some way. - The follow-up plan of this study based on the response level of the participants allowed us to obtain a strong response rate. - The responses obtained on participants' WTP may be sensitive/not robust when performed in a public health system context. - This is the first study concerning women's preferences related to SDM regarding BCSPs.
3. Please reformat the abstract so that it follows the structured abstract recommended in the journal's instructions for authors for research articles. See: https://bmjopen.bmj.com/pages/authors/#research	This section was rewritten: Objective: To analyse women's stated preferences for establishing the relative importance of each attribute of Shared Decision making (SDM) and their willingness to pay (WTP) for more participatory care in a breast cancer screening programme (BCSP). Design: A Discrete Choice Experiment was designed with 12 questions (choice tasks). It included three attributes: 'How the information is obtained' regarding benefits and harms, whether there is a 'Dialogue for mammography scheduled' between the healthcare professional and the woman, and, 'Who makes the decision' regarding participation in BCSP. The data were obtained using a survey that included 12 choice tasks, one question on WTP, and seven socioeconomic questions. The analysis was performed using conditional mixed-effects logit regression and stratification according to WTP.

	Setting: Data collection related to BCSP was conducted between June and November 2021 in Catalonia, Spain. Participants: Sixty-five women between 50 and 60 years old. Main outcome measures: Women's perceived utility of each attribute, trade-off on these attributes, and WTP for SDM in BCSP. Result: The only significant attribute was 'Who makes the decision'. The decision made by oneself alone (coefficient=2.879; 95% CI=2.297, 3.461) and the decision made together with a healthcare professional (2.375; 95%CI=1.573, 3.177) were the options preferred by women. The former contributes 21% more utility than the latter. Moreover, 52.3% of women stated a WTP 10 or more euros for SDM. Women's preferences regarding attributes did not influence their WTP. Conclusions: The women refused a current paternalistic model and prefer SDM or Informed Decision making for BCSP.
4. Please provide more detail on how the participants were recruited in the 2.4 Participants sections of your Methods. Was there a difference in methods of recruitment between participants who completed an online or a telephone survey?	4. Figure 2 was added on a recruitment and follow-up flow chart for the participants. The following sentence was also added: The women were invited to participate, voluntarily and unpaid, through a telephone call during which they were informed of the objectives of the research project and were asked to complete a survey via telephone or online if they agreed to participate. Those who did not complete the survey within one week were reminded (Figure 2).
5. Please discuss your low response rate as a limitation in your Discussion section.	In the first draft, we had miscalculated the ratio. This is because the total number of women invited to participate was considered, which is a mistake, since women who were invited to participate also received the survey should be considered. As confirmed in the article Watson, Verity; Becker, Frauke; de Bekker-Grob, Esther (2016). Discrete Choice Experiment Response Rates: A Meta-analysis. Health Economics, (), -. doi:10.1002/hec.3354

	According to, it was calculated on a base of 93 women, with the real ratio of 70.97% The correction was made in the article: Of the 292 women invited to participate, 93 accepted; 27 did not reply, and 66 submitted the survey (reflecting a 70.97% response rate), with one excluded due to a history of BC.
6. Please complete a thorough proofread of the text and correct any spelling and grammar errors that you identify. E.g. In figure 2 an error has been highlighted.	The article was initially edited in English by the ELSEVIER language service. We solicited a new correction.

VERSION 2 – REVIEW

REVIEWER	Hersch, Jolyn University of Sydney, School of Public Health
REVIEW RETURNED	26-Sep-2022

GENERAL COMMENTS	Strengths and Limitations bullet points are much better. But I don't really understand this sentence: "The follow-up plan of this study based on the response level of the participants allowed us to obtain a strong response rate." I think it might be better to delete this point, and instead put something about the fact that the study materials were carefully pilot-tested and refined through a two-stage piloting process. I am satisfied that the authors have addressed my other comments.
--

VERSION 2 – AUTHOR RESPONSE

Many thanks to the editor and Dr. Hersch for their last recommendations, the authors have noticed a great advance in the manuscript with the edits made. We hope that the changes in the manuscript will be necessary for publication in the journal.